# Deletion in the S1 Region of Porcine Epidemic Diarrhea Virus Reduces the Virulence and Influences the Virus-Neutralizing Activity of the Antibody Induced

**DOI:** 10.3390/v12121378

**Published:** 2020-12-02

**Authors:** Kuo-Jung Tsai, Ming-Chung Deng, Fun-In Wang, Shu-Hui Tsai, Chieh Chang, Chia-Yi Chang, Yu-Liang Huang

**Affiliations:** 1Animal Health Research Institute, Council of Agriculture, Executive Yuan, 376 Chung-Cheng Road, Tansui, New Taipei City 25158, Taiwan; krtsai@mail.nvri.gov.tw (K.-J.T.); mcdeng@mail.nvri.gov.tw (M.-C.D.); shtsai@mail.nvri.gov.tw (S.-H.T.); cchang@mail.nvri.gov.tw (C.C.); 2School of Veterinary Medicine, National Taiwan University, No. 1, Section 4, Roosevelt Road, Taipei 10617, Taiwan; fiwangvm@ntu.edu.tw

**Keywords:** PEDV, deletion, spike protein, virulence, neutralizing active

## Abstract

Porcine epidemic diarrhea virus (PEDV) causes severe diarrhea and a high rate of mortality in suckling pigs. The epidemic of PEDV that occurred after 2013 was caused by non-insertion and deletion of S gene (S-INDEL) PEDV strains. During this epidemic, a variant of the non-S-INDEL PEDV strain with a large deletion of 205 amino acids on the spike gene (5-17-V) was also found to co-exist with a non-S-INDEL PEDV without deletion (5-17-O). Herein, we describe the differences in the complete genome, distribution, virulence, and antigenicity between strain 5-17-O and variant strain 5-17-V. The deletion of 205 amino acids was primarily located in the S1^O^ domain and was associated with milder clinical signs and lower mortality in suckling pigs than those of the 5-17-O strain. The 5-17-V strain-induced antibody did not completely cross-neutralize the 5-17-O strain. In conclusion, the deletion in the S1 region reduces the virulence of PEDV and influences the virus-neutralizing activities of the antibody it induces.

## 1. Introduction

Porcine epidemic diarrhea (PED) is a swine disease that causes severe watery diarrhea and vomiting in pigs of all ages. The clinical signs in PED virus (PEDV)-infected pigs are age-dependent, wherein younger pigs have higher morbidity and mortality [1]. In weaned pigs, PEDV affects their innate immunity through a decrease in Peyer’s patch M cells and lysozyme expression in Paneth cells [2].

The PED virus is an enveloped, single-stranded, positive-sense RNA virus with the spike (S) protein on the envelope. It belongs to the order Nidovirale, family Coronaviridae, subfamily Coronavirinae, and genus *Alphacoronavirus* [3,4]. The genome of PEDV is approximately 28 kilobases encoding four non-structural proteins (ORF1a, ORF1b, ORF3a, and ORF3b) and four structural proteins (S, envelope (E), membrane (M), nucleocapsid (N)) [3,4]. The ORF3 protein has been associated with PEDV pathogenicity in that deletions between positions 245 and 295 of ORF3 decrease the virulence of PEDV [5]. The S protein binds to the aminopeptidase N receptors and is an important epitope of the neutralizing antibody [6,7,8,9]. The M protein plays important roles in viral assembly and the induction of interferon-alpha (IFN-α) [10]. The N protein induces cell-mediated immunity and inhibits IFN-β production during virus infection [11].

The PEDV strains are classified as genotypes 1 and 2 based on the S gene’s sequences. The strains in genotype 1, designated insertion and deletion of S gene (S-INDEL) of PEDV, were the predominant strains globally prior to 2013. These traditional PEDV strains were reported in England in 1971 [12], Belgium in 1978 [13], Japan in 1982 [14], Spain in 1985 [15], Korea in 1992 [16], China in 2004 [17], and in Thailand [18] and Taiwan [19] in 2007. In 2010, a novel mutated strain of PEDV was found in China, classified as genotype 2, and designated non-S-INDEL PEDV [20,21]. Later, this novel virus was transmitted rapidly throughout countries around the Pacific Ocean including the United States of America (USA), Japan, South Korea, Taiwan, Thailand, Vietnam, Mexico, Peru, the Dominican Republic, Columbia, and Canada in 2013 and 2014 [14,19,20,21,22,23,24]. It was estimated that over 7 million suckling pigs were lost in the USA during the one-year epidemic [25]. Comparison of the S protein sequences of the S-INDEL PEDV strain revealed that the novel PEDV strain had a 4-amino acid (aa) insertion between positions 55 and 56, a 1-aa insertion between positions 135 and 136, and a 2-aa deletion between positions 158 and 159 [19]. Virulence studies have demonstrated that it is highly pathogenic to suckling pigs [1,26].

In late 2013, the PED outbreak that occurred in Taiwan was also caused by the novel PEDV. Subsequently, two novel PEDV strains (strains 5-17-O and 5-17-V) were isolated from the same diarrhea samples. Comparison of the S genes showed that the 5-17-V strain had a 205-aa deletion in the S1 region. This study was carried out to understand the effects of this deletion on the virus distribution in herds, virulence, and antigenicity between the two novel PEDV strains.

## 2. Materials and Methods

### 2.1. Viral Isolation

The Vero cell line was used to isolate PEDV from swine diarrhea samples using the protocols of Chen (2014) [27]. Briefly, the diarrhea samples were prepared in 10% emulsion in post-inoculation medium (PI) containing minimum essential medium (MEM) supplemented with 0.02% yeast extract, 0.3% tryptose broth, and trypsin 250 (10 μg/mL). The monolayer Vero cells were inoculated with the emulsion sample at 37 °C with 5% CO_2_ for 2 h. After the inoculation, the PI was added, and the mixture was incubated at 37 °C with 5% CO_2_. When the cytopathic effect (CPE) of cell fusion was present, CPE was confirmed by indirect fluorescent-antibody assay (IFA) using PEDV mAb (DE1-1a clone, from the Animal Health Research Institute, Taiwan).

### 2.2. RNA Extraction

The viral RNA of PEDV was extracted from the culture supernatant and tissue emulsion using the QIAamp^®^ Viral RNA Mini kit following the manufacturer’s instructions (Qiagen, Hilden, Germany).

### 2.3. Sequencing and Analysis of PEDV’s Complete Genome

The two novel strains (5-17-O and 5-17-V) and one traditional strain (4-2) were sequenced to obtain the complete genome using the methods of Huang (2013) [21] and Chen (2014) [27]. A total of seven fragments of each strain were amplified by reverse-transcription polymerase chain reaction (RT-PCR) using PEDV-specific primer, superscript^®^ III, and high-fidelity Platinum^®^ Taq DNA polymerase. Following the amplification, the product of RT-PCR was gel purified and sequenced by next-generation sequencing with the Illumina MiSeq™ and TruSeq Nano DNA Library Prep Kit. Finally, the sequences of the full genomes were constructed in CLC Genomics Workbench software and mapped to reference the PEDV strain USA/Indiana/17846/2013 (accession number: KF452323). The full genome of the 5-17-O (MW165327), 5-17-V (MW165328), and 4-2 (MW165329) strains were submitted to the GenBank of the National Center for Biotechnology Information (NCBI). The nucleotide (nt) sequences of the 5′UTR, ORF1a, ORF1b, S, ORF3, E, M, N, and 3′UTR genes of the reference strains (traditional Taiwanese PEDV: 4-2 strain; S-INDEL strain: CV777; non-S-INDEL: USA/Indiana/17846/2013) and two novel PEDV strains (5-17-O and 5-17-V) were compared. Furthermore, the multiplex alignment of the S genes of Taiwanese PEDV strains and PEDV reference strains obtained from GenBank of NCBI was analyzed in DNASTAR software using the Clustal W method. Phylogenetic analysis of the aligned sequences was performed by the neighbor-joining method and by bootstrap analysis with 1000 replicates in the DNASTAR software (DNASTAR Inc., Madison, WI, USA).

### 2.4. Distribution of Two Novel PEDV Strains in the Farms of Taiwan

From the 2013 epidemic, two novel PEDV strains (5-17-O and 5-17-V) were isolated from diarrhea samples. To understand the population distribution of these two novel PEDV strains in farms, a total of 105 diarrhea samples from 64 farms (herds) were collected in the year 2014, and the strains were detected by differential diagnostic RT-PCR. Based on the sequences of the S genes of the two novel PEDV strains, universal primers (SF1 and SR739) for PEDV were designed. The primer sequences of SF1 and SR739 were 5′-TGCTAGTGCGTAATAATGAC-3′ and 5′-CCTTCTGGTATGTGGCCATT-3′, respectively. The total volume of the reaction, 25 μL, contained 2.5 μL extracted RNA, 1× DNA polymerase buffer, 4 units of recombinant RNase inhibitor (Promega, Madison, WI, USA), 1 unit of AMV reverse transcriptase (Promega, Madison, WI, USA), 1 unit of GoTaq^®^ Flexi DNA polymerase (Promega, Madison, WI, USA), 0.2 μM of deoxyNTP mixture, and 0.4 μM of each primer (SF1 and SR739). The reaction conditions involved reverse transcription at 42 °C for 40 min and amplification with 1 cycle at 95 °C for 5 min, 35 cycles of denaturation at 95 °C for 30 s, annealing at 55 °C for 30 s, extension at 72 °C for 60 s, and 1 cycle of final extension at 72 °C for 10 min. Finally, the products of the RT-PCR were electrophoresed in agarose gel. The predicted RT-PCR amplified products for the 5-17-O and 5-17-V strains were 802 bp and 187 bp, respectively.

### 2.5. Growth Curves of Two Novel PEDV Strains in the Various Concentration of Trypsin

To study the effect of trypsin concentration on the replication of novel PEDV strains, the kinetics of PEDV replication were determined. First, PIs with 0.1, 1, 5, and 10 μg/mL trypsin were prepared. Monolayer Vero cells in the 6-well plates were inoculated with 0.1 multiplicities of infection (MOI) of 5-17-O or 5-17-V strains at 37 °C for 2 h. Afterward, the inoculants were removed by washing with PI and replaced with fresh PI. The culture supernatant and cells were harvested at 3 h intervals between 0 and 27 h post-inoculation (HPI), and the PEDV titers in 50% tissue culture infectious doses (TCID_50_) were determined.

### 2.6. Characterization of the Virulence In Vivo

To compare the virulence of the 5-17-O and 5-17-V strains, an animal experiment approved by the Institution Animal Care and Use Committee (IACUC) of the Animal Health Research Institute (AHRI Approval number: A04006) was conducted. Thirty-six PEDV-seronegative, one-week-old, healthy piglets were randomly assigned to 9 groups of 4 and housed separately in 9 rooms. The piglets in groups 1–4 were inoculated orally with 10^3^, 10^4^, 10^5^, and 10^6^ TCID_50_ of the 5-17-O strain, respectively. Those in groups 5–8 were inoculated orally with 10^4^, 10^5^, 10^6^, and 10^7^ TCID_50_ of the 5-17-V strain, respectively. Those in group 9 were negative controls inoculated with MEM. After inoculation, the clinical signs, including diarrhea, weight, and body temperature, were recorded during the experimental period. Based on the fecal formation, diarrhea was scored as follows: 0 = solid; 1 = soft to pasty; 2 = semi-liquid; 3 = complete liquid. The feces and serum were collected, respectively, daily and at two-day intervals at 0–11 days post-inoculation (DPI). The piglets that died during the experimental period were necropsied. The surviving piglets were necropsied at 14 DPI. The viral loads were detected by real-time reverse-transcription PCR (RRT-PCR). The antibody against PEDV was detected by IFA and a virus-neutralizing (VN) assay.

### 2.7. Detection of PEDV Loads by RRT-PCR

The PEDV loads in the feces were detected by RRT-PCR. The primer and probe in the RRT-PCR of PEDV were designed to target the nucleoprotein gene. The sequences of the primer and probe were 5′-GCTTCTCAGAACAGAGGA-3′, 5′-CATCGCGTGATGTTACAC-3′, and 5′-FAM-CAATAACAAGTCTCGTAACCAGTCCAA-BHQ1-3′. The total volume of 20 μL contained 3 μL of extracted nuclear acids, 1× Kappa Probe Fast qPCR Master mix (KAPA biosystems, Boston, MA, USA), 40 units of SuperScript^TM^ III Reverse Transcriptase (Invitrogen, Carlsbad, CA, USA), 0.3 μM of PEDV probe, and 0.5 μM of each PEDV primer. The reaction conditions involved initial incubation at 42 °C for 30 min and 94 °C for 5 min followed by 45 cycles of denaturation at 95 °C for 20 s, annealing at 60 °C for 30 s, and extension at 72 °C for 30 s. Finally, the threshold line was set at a fluorescence level of 5.

### 2.8. Detection of Anti-PEDV Antibody

The anti-PEDV antibody in pig serum was detected by IFA. Briefly, Vero cells in 96-well plates were inoculated with 200 TCID_50_ PEDV at 37 °C with 5% CO_2_ for 16 h. The supernatant was disposed, and the cells were dried at 37 °C for 1 h before being fixed with 10% neutral-buffered formalin at room temperature for 10 min and then washed 3x with PBS. The two-fold serially diluted serum sample was added onto wells at 37 °C for 1 h and then washed again. Then FITC-labeled goat-anti-swine IgG (Jackson ImmunoResearch Inc., West Grove, PA, USA) was added, and the mixture was held at 37 °C for 1 h, washed again, and observed under fluorescence microscopy.

### 2.9. Virus-Neutralizing Assay for PEDV

Anti-PEDV-neutralizing antibody in serum was determined in Vero cells using 5-71-V or 5-17-O strains. In a separate 96-well plate, two-fold serially diluted (from 8 to 512 fold) serum samples were mixed with 200 TCID_50_/0.1 mL PEDV (strain 5-17-O or 5-17-V) at 37 °C with 5% CO_2_ for 2 h. The pre-seeded Vero cells were washed 2 times with phosphate-buffered saline (PBS) before the mixture was transferred and incubated at 37 °C with 5% CO_2_ for 2 h. The mixture was removed from the cells and washed again with PBS before being placed in fresh PI medium and incubated at 37 °C with 5% CO_2_ for 2 days. The cutoff was based on the fusion CPE.

### 2.10. Cross-Neutralization in Serum between the 5-17-O and 5-17-V Strains

Due to the fact that the PEDV’s VN antibodies were negative in the sera (see Section 2.6), a total of 66 anti-PEDV sera from the other pig experiments of inactivated 5-17-O and 5-17-V vaccines were collected, comprising 38 samples from 5-17-O vaccinated piglets and 28 samples from 5-17-V vaccinated piglets. The anti-PEDV-neutralizing antibodies in 66 sera were all detected against the 5-17-O and 5-17-V strains.

### 2.11. Statistical Analysis

The Student’s *t*-test, carried out in Microsoft Office Excel 2007, was used to compare the differences between the two groups. The comparisons of the groups were analyzed by one-way analysis of variance (ANOVA). The ANOVA combined with Duncan’s multiple range test was performed in SAS for Windows 6.12 (SAS Institute Inc., Cary, NC, USA). A *p*-value < 0.05 was considered significant.

## 3. Results

### 3.1. Comparison of Genetic Similarity between 5-17-O and 5-17-V Strains of PEDV

Differences in protein sequences were found in the ORF1 and S proteins of the two strains, whereas the sequences of other proteins, including 5′UTR, ORF3, E, M, N, and 3′UTR, were completely the same. Strain 5-17-V had one amino acid difference in position 1652 of ORF1a, from A of the 5-17-O strain to V of 5-17-V.

In the S protein, 5-17-V had a deletion of 205 amino acids (Figure 1) and a difference of three amino acids from the 5-17-O strain. The deletion was from the aligned positions 23 to 229 of the amino acids. The three changes from the 5-17-O to the 5-17-V strains were from S to G, from P to L, and from P to Q in the aligned positions 254, 506, and 1063, respectively. From the comparison of the deleted locations among the variant non-S-INDEL strains (5-17-V, LC022792, and KM392229), the same gap was located between the aligned positions 34 and 220. Based on the same gap, the gap of each strain extended several amino acids.

### 3.2. Comparison of Difference of Various Genes between Reference PEDV and Novel Taiwanese PEDV

Phylogenetic analysis of the S proteins revealed that the 5-17-O and 5-17-V strains were both classified as genotype 2, similar to those of non-S-INDEL strains in the USA, Japan, South Korea, and China (Figure 2). The similarity of the S gene between 5-17-O and the other non-S INDEL strains was from 98% to 100%. The similarity of the S gene between 5-17-V and the variable non-S INDEL strains (LC022792 and KM 392229) was from 99.3% to 99.7%. On the other hand, the traditional Taiwanese PEDV strain, 4-2, was classified as group 1 (S-INDEL) and had a similarity of 97.8% to the S gene from KP768390 (Taiwan/HC070225/2007).

The similarity of nucleotides and amino acids in various genes between the reference strains and the two novel Taiwanese PEDV strains (5-17-V and 5-17-O) were compared, and the results of the 5-17-V and 5-17-O strains were the same for the 5′UTR, ORF1a, ORF1b, ORF3, E, M, N, and 3′UTR genes; the similarity of nucleotides was between 99.6% and 100% for the non-S-INDEL strain (KF452323) and between 96.7% and 99.4% for the S-INDEL strain (MW165329 and AF353511) (Table 1). However, a deletion of 28 amino acids in the ORF1a gene was found in the 4-2 strain, located between the aligned positions 1011 and 1038. The similarity of the S proteins between the 5-17-O and S-INDEL strains was from 93.6% to 95.2%. The difference between 5-17-O and 4-2 strains was the same as the pattern between S-INDEL and non-S-INDEL strains in that 5-17-O had a 4-aa insertion between aligned positions 59 and 562, a 1-aa insertion in aligned position 140, and a 2-aa deletion between aligned positions 161 and 162 (Figure 1).

### 3.3. Population Distribution of the 5-17-O and 5-17-V Strains in Taiwan

A total of 105 diarrhea samples from 64 farms were examined by differential diagnostic RT-PCR of PEDV. The results showed that 73 samples from 46 farms had 802 bp of RT-PCR product and belonged to the 5-17-O strain. In the 5-17-O positive samples, 10 samples from six farms also had 187 bp of RT-PCR product and belonged to the 5-17-V strain. The other 32 samples were negative according to the differential diagnostic RT-PCR of PEDV.

### 3.4. Comparison of Replication Kinetics between the 5-17-O and 5-17-V Strains in an In Vitro Experiment

The replication kinetics of the two novel PEDV strains in various concentrations of trypsin were determined by an in vitro study. The growth curves of the two PEDV strains in each condition were established and compared to each other at the same time (Figure 3). The growth curves of the two PEDV strains in the same condition did not differ significantly. The PEDV titer in each condition did not significantly increase in the early 9 HPI. From 9 to 18 HPI, the titers of the 5-17-O and 5-17-V strains in the high trypsin concentrations (5 and 10 μg/mL) increased exponentially from 10^4.9^ to 10^6.85^ TCID_50_/mL of 5-17-O/5, 10^4.84^ to 10^6.65^ TCID_50_/mL of 5-17-O/10, 10^5.15^ to 10^6.62^ TCID_50_/mL of 5-17-V/5, and 10^5.02^ to 10^6.79^ TCID_50_/mL of 5-17-V/10. The titers of the 5-17-V and 5-17-O strains in the high trypsin concentrations were significantly higher than that of the low trypsin concentrations (0.1 and 1 μg/mL) at 12, 15, and 18 HPI. After 18 HPI, the titers of the two PEDV strains in the high trypsin concentrations gradually decreased. In the low trypsin concentration, the titer of the 5-17-O strain slowly increased from 10^4.5^ to 10^5.81^ TCID_50_/mL of 5-17-O/0.1 and from 10^4.54^ to 10^5.81^ TCID_50_/mL of 5-17-O/1 at 9 to 27 HPI, and the peak titer was at 27 HPI. However, the lower titer was revealed in the 5-17-V/0.1 and 5-17-V/1. The peak titers of the 5-17-V strain in 0.1 and 1 μg/mL trypsin were only 10^4.9^ TCID_50_/mL at 15 HPI and 10^5.5^ TCID_50_/mL at 18 HPI, respectively.

### 3.5. Comparison of Virulence between the 5-17-O and 5-17-V Strains In Vivo

#### 3.5.1. Clinical Signs

The piglets in the control group were all healthy during the experimental period. The piglets inoculated with PEDV all exhibited diarrhea within 2 DPI, which continued for 3–9 DPI. The averaged diarrhea day (period) was affected by the inoculated dosages, wherein those inoculated with 10^6^ and 10^7^ TCID_50_ of 5-17-V (groups 7 and 8) showed significantly more diarrhea days than those with 10^4^ and 10^5^ TCID_50_ of 5-17-V (groups 5 and 6) (Table 2). No significant difference in diarrhea days was found in the 5-17-O strain group (groups 1–4). To compare the extent of the diarrhea, more severe diarrhea scores were found at 3–5 DPI in the piglets with 10^5^ or 10^6^ TCID_50_ of 5-17-O (groups 3 and 4) or 10^7^ TCID_50_ of 5-17-V (group 8) than in the other groups (Figure 4). Comparing the diarrhea scores in low-dose inoculated piglets between 5-17-O and 5-17-V, piglets with 10^3^ or 10^4^ TCID_50_ of 5-17-O (groups 1 and 2) also had more severe diarrhea than piglets with 10^4^ or 10^5^ TCID_50_ of 5-17-V (groups 5 and 6) at 3–4 DPI. During the experimental period, 50% mortality occurred in piglets inoculated with 10^6^ TCID_50_ of 5-17-O (group 4) and 10^7^ TCID_50_ of 5-17-V (group 8) (Table 2). Two piglets in group 4 (5-17-O/10^6^) died separately at 5 and 6 DPI. Two piglets in group 8 (5-17-V/10^7^) died separately at 6 and 8 DPI. Before the piglets’ death, they showed continual severe diarrhea for more than three days.

#### 3.5.2. Viral Shedding in Feces

No PEDV was detected in the feces of piglets in the negative controls (group 9) during the experimental period. The PEDV shedding from feces was found in all piglets inoculated with PEDV (Table 2). In piglets inoculated with the 5-17-O strain (groups 1–4), PEDV shedding was detected within 2 DPI and continued to 6 DPI. After 7 DPI, some piglets recovered gradually, but in each group, one or more piglets continuously shed PEDV in feces until the end of the experiment. The earliest PEDV shedding time was detected at 1 DPI in piglets inoculated with 10^7^ TCID_50_ of the 5-17-V strain (group 8) and 10^6^ TCID_50_ of the 5-17-O strain (group 4). Comparison of average shedding days between the 5-17-O and 5-17-V strains showed no significant difference (Table 2).

#### 3.5.3. The Titer of Anti-PEDV Serum Antibody Detected by IFA and Neutralizing Assay in Experimentally Infected Pigs

The anti-PEDV antibody was not detected in any piglets by IFA before the in vivo experiment. At 6 DPI, anti-PEDV antibodies were detected by IFA in the piglets in the 5-17-V inoculation (groups 5–8), and the titer gradually increased from 6 to 10 DPI. However, in 5-17-O inoculated piglets (groups 1–4), only two piglets with inoculation of 10^6^ TCID_50_ (group 8) had anti-PEDV antibodies at 10 DPI. However, no neutralizing antibodies against the 5-17-V and 5-17-O strains were detected in the sera of all piglets at 10 DPI.

### 3.6. Cross-Neutralization between the 5-17-O and 5-17-V Strains

As the VN antibodies of the pig sera were negative, additional sera from the other piglets immunized with inactivated 5-17-O or 5-17-V vaccines were collected to assay for the cross-neutralization between 5-17-O and 5-17-V. Based on the serum origin and anti-homologous PEDV neutralizing antibodies, a total of 66 sera were divided into 23 samples of 5-17-O low (4 to 32 fold), 15 samples of 5-17-O high (64 to 512 fold), 17 samples of 5-17-V low (8 to 32 fold), and 11 samples of 5-17-V high (64 to 512 fold). The VN titers against the strain 5-17-V in sera of the 5-17-O low, 5-17-V low, and 5-17-V high groups were all equal/similar or higher than that against strain 5-17-O. The ratio of the VN titers against the 5-17-V strain were over four-fold higher than that of the VN titer against the 5-17-O strain: 30.4% (7/23), 47.1% (8/17), and 72.7% (8/11) in the 5-17-O low; 5-17-V low and 5-17-V high, respectively. The neutralizing titers against 5-17-V strains in the 5-17-O low (5.0 ± 2.3 log2), 5-17-V low (2.9 ± 1.6 log2), and 5-17-V high (6.7 ± 1.0 log2) samples were, significantly higher than those against 5-17-O (3.6.0 ± 1.1 log2 of 5-17-O low; 2.9 ± 1.6 log2 of 5-17-V low; 4.1 ± 1.6 log2 of 5-17-V high, respectively) (Figure 5). However, those in the 5-17-O high sample were not significantly different in the VN assay against the 5-17-V and 5-17-O strains.

## 4. Discussion

The S protein is the target gene for evaluating the evolution of PEDV and the origin of PEDV transmission. The non-S-INDEL PEDV, with insertion and deletion in the S protein, has rapidly spread globally and possibly originated from the AH2012 strain [21,27]. In the 2013 PED epidemic, when non-S-INDEL PEDV first emerged in Taiwan [19], we found a variant of the non-S-INDEL PEDV strain (5-17-V) with a deletion of 205 amino acids in the S1 region during the routine diagnosis of swine diseases. The PEDV with large deletions in the S1 region was also found in Japan (deletion of 194 amino acids) [28] and the USA (deletions of 194 to 204 amino acids) [29]. Nevertheless, the size and locations of the deletions in the S1 region differed slightly among these various PEDV strains. The deletion size ranged from 194 to 205 amino acids, and the segment between residues 34 and 212 of the S protein was completely absent. The deletion of 205 amino acids also occurred in the 5-17-V strain and JKa-295/CS1de205 (KU363101) [30] but in different locations. The variant non-INDEL PEDV strain, 5-17-V, was first found in Taiwan.

The phylogenetic analysis of the S gene revealed that 5-17-O was a Taiwanese non-S-INDEL PEDV strain. The inserted and deleted patterns and sequences in the non-S-INDEL found in Taiwan were highly similar to those of the non-S-INDEL strains found in the USA, Japan, South Korea, and China and was different from that of the traditional Taiwanese PEDV strain. These results indicated that the Taiwanese non-S-INDEL was, in fact, a new invasive strain. The same discourse was demonstrated in a recent study showing that the Taiwanese non-S-INDEL PEDV strains were highly related with the non-S-INDEL strain in the USA [19]. In addition, the variable non-S-INDEL strain, similar to the 5-17-V strain with a large deletion, was reported in the USA and Japan. Our investigation found that the 5-17-V and 5-17-O usually co-existed in the same sample. Therefore, the piglets might be simultaneously infected with both the 5-17-V and 5-17-O strains, suggesting that the 5-17-V invasion followed that of the 5-17-O invasion at the same stage.

Viral evolution allows viruses to adapt to changes in host environments, leading to the development of various strains. The pathways of viral evolution include mutation, insertion, deletion, and recombination of genes. The non-S-INDEL PEDV strain is a new PEDV strain of genotype 2 with mutation, insertion, and deletion as compared with the traditional PEDV strain of genotype 1. In this study, we found that the variant 5-17-V strain coexisted in 14% of cases (10/73 samples, see Section 3.3) of non-S-INDEL PEDV infection. This large deletion of the S protein has also occurred in the evolutions of other coronaviruses such as transmissible gastroenteritis virus (TGEV) [31] and Middle East respiratory syndrome coronavirus [32]. The gene deletion of TGEV led to a change in the virus tropism from intestinal enterocytes to respiratory tissues, indicating that the deletion of the S protein is one pathway in the process of coronavirus evolution.

Trypsin plays an important role during PEDV replication. Recent studies and this study demonstrate that insufficient load and lack of trypsin decrease the efficacy of PEDV production in cells [33,34]. Furthermore, we also found that the replication of the 5-17-V strain was significantly more trypsin dependent than the 5-17-O strain. The difference in trypsin dependence between these two strains is presumed to be associated with the deletion of the PEDV S protein. Strain 5-17-V had the deletion of 205 amino acids of the S protein located mainly in the S1^O^ domain and partially in the S1^A^ domain. During PEDV infection, trypsin can cleave the S protein into S1 and S2 subunits, leading the S1 subunit to bind to the aminopeptidase N receptor [33,34]. If the S1 subunit is thus incomplete, it could possibly lower the binding capability of PEDV. In addition, trypsin is able to enhance the viral budding and fusion formation of CPE in infected cells [33,34]. The effect of both a lower trypsin concentration and an incomplete S protein of PEDV could significantly decrease the efficacy of PEDV replication.

The S protein is one of the determinants of virulence in the pathogenesis of PEDV [35], wherein high virulence in the suckling pigs occurred with the non-S-INDEL PEDV strains. When the S1 region of S-INDEL PEDV strains was experimentally replaced with that of non-S-INDEL strains, more severe clinical signs were observed in suckling pigs infected with the recombined strain [33]. In this study, we demonstrated milder clinical signs and lower mortality in suckling pigs with 5-17-V infection, indicating that the deletion of 205 amino acids decreased the virulence of PEDV. The variant non-S-INDEL (5-17-V), a low-virulent PEDV strain, has the advantage of easier attenuation, but its antigenicity to induce the antibody production is not completely characterized. For the development of a vaccine candidate, the reduced antigenicity of the variant 5-17-V strain would be improved by an adjuvant to further induce the adaptive immunity. In addition, higher infection doses induced more severe diarrhea and higher mortality in suckling pigs, as found in other studies [36,37], implying that the infectious dose directly influences the PEDV pathogenesis.

The receptor-binding domain of PEDV, which resides in the S protein, is also associated with the induction of a neutralizing antibody [7,8]. The S1^B^ domain of PEDV attaches to the porcine aminopeptidase N (pAPN) receptor during virus entry [9,38]. The S1^O^ domain, which binds to Sia, is associated with the hemagglutinating activities of PEDV [39,40]. The large deletion of S in the 5-17-V strain, located mainly in the S1^O^ domain and partially in the S1^A^ domain, induced a neutralizing antibody that has a lower capacity to cross-neutralize 5-17-O (Figure 5), giving the virus a survival advantage. The neutralizing epitopes of the S1^O^ domain were also determined [41], suggesting that positions 23–227 of the S protein contain the potent neutralizing epitopes, and this deletion affected the neutralizing activity of PEDV-induced serum.

## 5. Conclusions

In conclusion, the variant PEDV (5-17-V) with a large deletion in S was first found in Taiwan and commonly coexists in herds infected with PEDV without a large deletion in S. This large deletion, primarily located in the S1^O^ domain, is associated with the reduced virulence of PEDV in vivo and influences the neutralizing activities of the antibody induced by the variant PEDV to cross-neutralize those PEDVs without a large deletion on S, providing the virus a survival advantage.

## Figures and Tables

**Figure 1 viruses-12-01378-f001:**
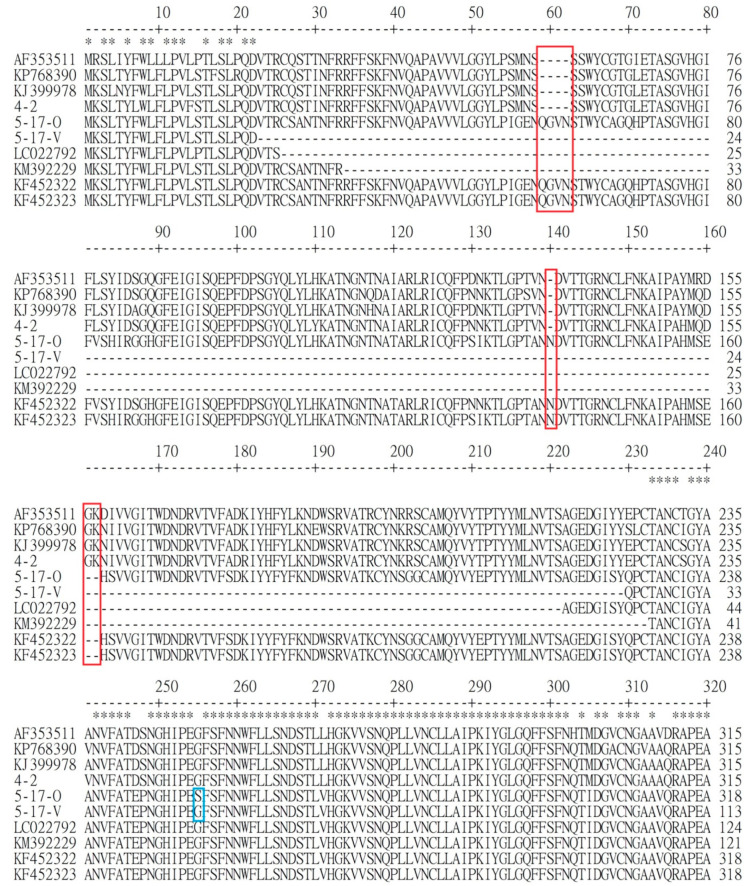
The alignment of the N-terminal 1–320 amino acids in the S protein of the 5-17-O, 5-17-V, 4-2, and porcine epidemic diarrhea virus (PEDV) reference strains. The reference strains consisted of insertion and deletion of S gene (S-INDEL) strains (AF353511, KJ399978, and KP768390), variable non-S-INDEL strains (LC022792 and KM392229), and non-S-INDEL strains (FK452323 and KF452322). The 5-17-O and 5-17-V were newly invaded Taiwanese PEDV strains post-2013, and their inserted and deleted patterns were similar to the non-S-INDEL PEDV genotype strains. The 4-2 strain was the traditional Taiwanese PEDV strain, and its inserted and deleted patterns were the same as the KP768390 (Taiwan/HC070225/2007) and the other S-INDEL PEDV genotype strains (AF353511 and KJ399978). The stars indicate the same amino acid in all strains. The red boxes reveal the inserted and deleted regions between non-S-INDEL and S-INDEL strains. The blue box shows the difference in amino acids between 5-17-O and 5-17-V.

**Figure 2 viruses-12-01378-f002:**
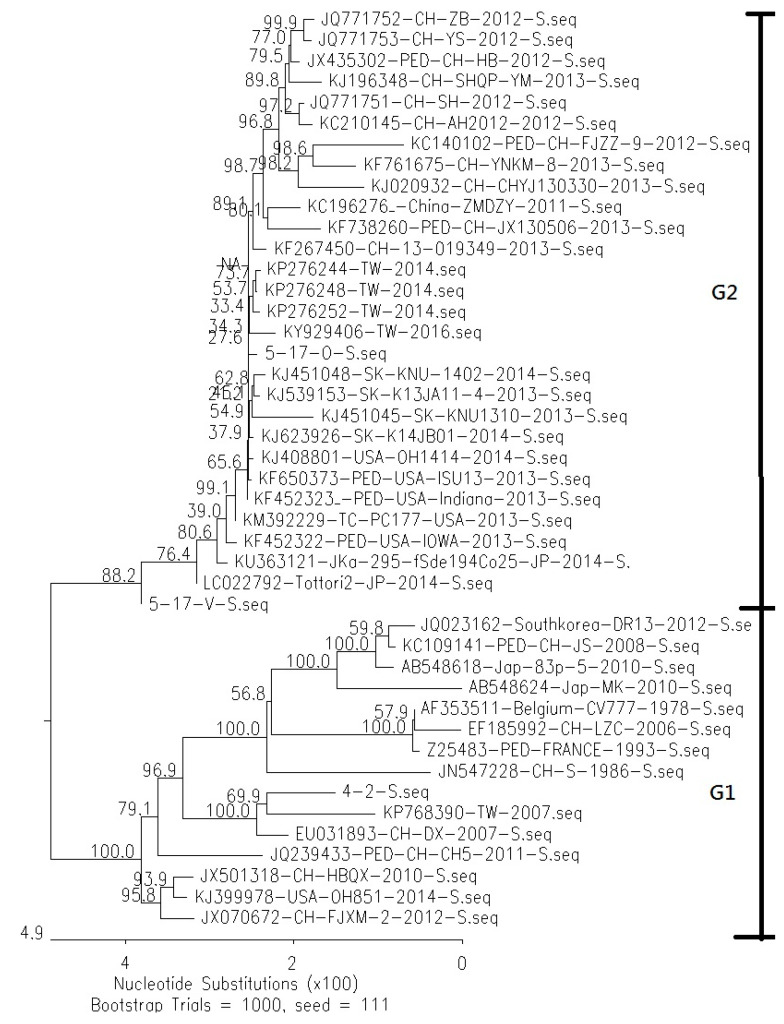
The results of phylogenetic analysis in the spike genes between Taiwanese PEDV and reference PEDV strains are shown. The sequence of the S gene in the PEDVs was aligned and performed in the DNASTAR software (version 7.1.0) using the neighbor-joining methodology with 1000 replicates in bootstrapping analysis. The 5-17-O, 5-17-V, and non-S-INDEL PEDV strains were classified as group 2 (G2). However, the traditional Taiwanese PEDV strains (4-2 and KP768390) and the S-INDEL PEDV strains were classified as group 1 (G1).

**Figure 3 viruses-12-01378-f003:**
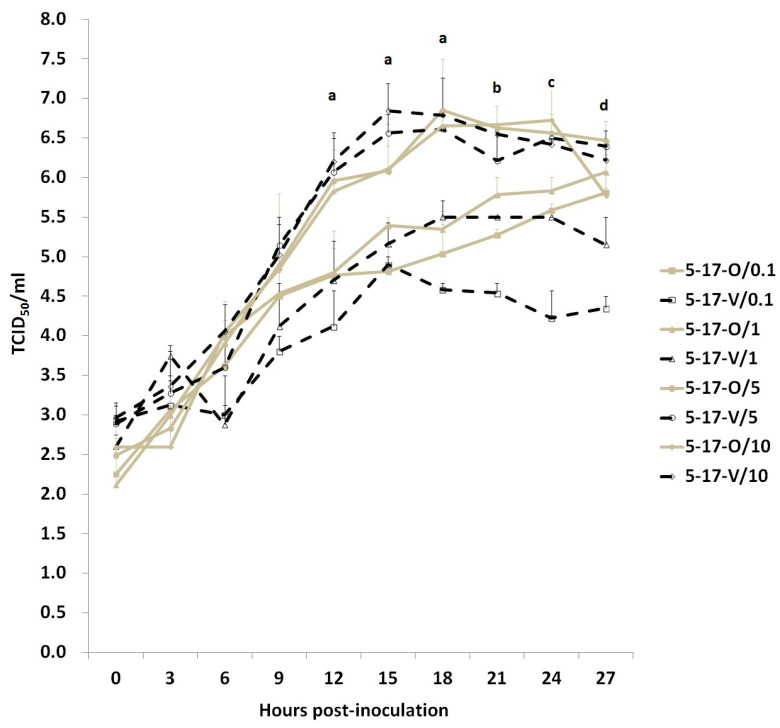
The growth curves of the 5-17-O and 5-17-V strains among various trypsin concentrations (0.1–10 μg/mL) of post-inoculation medium (PI) were determined in Vero cells. The solid and gray lines represent the 5-17-O strain. The dotted and black lines represent the 5-17-V strain. The square, triangle, ring, and diamond symbols represent 0.1, 1, 5, and 10 μg/mL of trypsin, respectively. ^a^ Indicates that the titers of 5-17-V and 5-17-O in the high trypsin concentrations (5 and 10 μg/mL) were significantly higher than that of the low trypsin concentrations (0.1 and 1 μg/mL). ^b^ Indicates that 5-17-V/10, 5-17-O/5, and 5-17-O/10 had significantly higher titers than 5-17-V and 5-17-O in the low trypsin concentrations. ^c^ Indicates that 5-17-O in the high trypsin concentrations had a significantly higher titer than 5-17-V and 5-17-O in the low trypsin concentrations. ^d^ Indicates that 5-17-O/5, 5-17-V/5, and 5-17-V/10 had significantly higher titers than 5-17-V/0.1 and 5-17-V/1.

**Figure 4 viruses-12-01378-f004:**
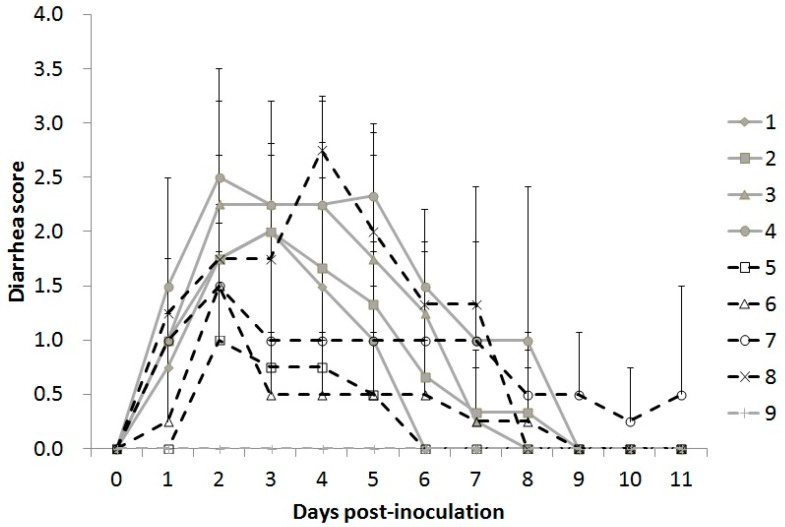
The averaged diarrhea scores of each group during the experimental period. The piglets in groups 1 to 4 were inoculated with 10^3^ to 10^6^ TCID_50_ of 5-17-O (solid and gray lines), respectively. The piglets in groups 5 to 8 were inoculated with 10^4^ to 10^7^ TCID_50_ of 5-17-V (dotted and black lines), respectively. Group 9 was the negative control. The diarrhea was scored as follows: 0 = solid; 1 = soft to pasty; 2 = semi-liquid; 3 = complete liquid.

**Figure 5 viruses-12-01378-f005:**
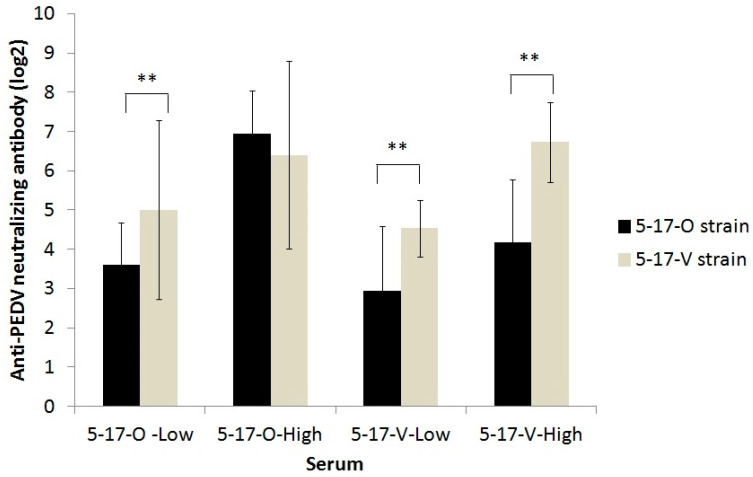
Cross-neutralizing antibody against the 5-17-O and 5-17-V strains between the sera of 5-17-O and 5-17-V vaccinated piglets. Based on the serum origin and anti-homologous PEDV neutralizing antibodies, a total of 66 sera were divided into 23 samples of 5-17-O low (4 to 32 fold), 15 samples of 5-17-O high (64 to 512 fold), 17 samples of 5-17-V low (8 to 32 fold), and 11 samples of 5-17-V high (64 to 512 fold). ** Indicates a significant difference (*p* < 0.01).

**Table 1 viruses-12-01378-t001:** The similarity of nucleotides and amino acids in the various genes between reference PEDV strains and novel Taiwanese PEDV strains.

Strains	Gene	4-2 ^c^ (MW165329)		CV777 (AF353511)		USA/Indiana/17846/2013 (KF452323)	
		Nucleotide	Amino Acids	Nucleotide	Amino Acids	Nucleotide	Amino Acids
5-17-O (5-17-V)	5′-UTR	98.9 ^a^	-	98.9	-	100	-
	ORF1a	98.6 ^b^	98.7	97.7	97.7	99.9	99.9
	ORF1b	98.7	98.4	97.8	99.3	100	100
	ORF3	96.7	96.4	96.7	96.0	99.6	99.1
	S	94.9 (96.9)	95.2 (97.6)	94.0 (94.7)	93.6 (95.6)	99.9 (99.0)	99.8 (99.7)
	E	97.4	98.7	97.0	98.7	100	100
	M	98.1	99.6	98.2	98.7	100	100
	N	98.6	99.3	96.0	97.1	100	100
	3′-UTR	99.4	-	97.9	-	100	-

^a^ The unit of similarity for nucleotides and amino acids is percentage (%). ^b^ The deletion of 28 amino acids in the ORF1a region of the 4-2 strain was located between the aligned positions 1011 and 1038. ^c^ Strains 4-2 and CV777 belonged to the S-INDEL strain. Strain USA/Indiana/17846/2013 belonged to the non-S-INDEL PEDV strain.

**Table 2 viruses-12-01378-t002:** The clinical signs of piglets inoculated with various PEDV dosages.

Group	Strain/Inoculated Load (TCID_50_)	Diarrhea%/Shedding%	Average Diarrhea Days	Average Shedding Days	Mortality (%)
1	5-17-O/10^3^	100/100	4.5 ± 0.9 ^c,d^	7.8 ± 1.5 ^b^	0
2	5-17-O/10^4^	100/100	5.0 ± 1.9 ^c,d^	8.3 ± 1.1 ^b^	0
3	5-17-O/10^5^	100/100	6.0 ± 0.7 ^c,d^	8.3 ± 1.1 ^b^	0
4	5-17-O/10^6^	100/100	6.0 ± 1.4 ^c,d^	8.5 ± 2.9 ^b^	50 (2/4) *
5	5-17-V/10^4^	100/100	1.5 ± 0.9 ^a,b^	6.3 ± 0.8 ^b^	0
6	5-17-V/10^5^	100/100	3.5 ± 2.6 ^b,c^	7.5 ± 0.5 ^b^	0
7	5-17-V/10^6^	100/100	9.0 ± 2.4 ^e^	7.3 ± 0.4 ^b^	0
8	5-17-V/10^7^	100/100	7.3 ± 2.0 ^d,e^	6.3 ± 1.4 ^b^	50 (2/4) **
9	MEM	0	0.0 ± 0.0 ^a^	0.0 ± 0.0 ^a^	0

* Two piglets in group 4 died separately at 5 and 6 days post-inoculation (DPI). ** Two piglets in group 8 died separately at 6 and 8 DPI. ^a–e^ Values with different superscripts indicate that the differences among groups are statistically significant (*p* < 0.05). TCID_50_: 50% tissue culture infectious doses.

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
