# Peer review of "Deletion in the S1 Region of Porcine Epidemic Diarrhea Virus Reduces the Virulence and Influences the Virus-Neutralizing Activity of the Antibody Induced"

_viruses, 2020, doi:10.3390/v12121378_

Round 1

Reviewer 1 Report

The authors submitted a manuscript (viruses-981116) describing results about the impact of a deletion in the spike protein of PEDV on virus replication in vitro as well as the outcome of virus-caused pathology in an animal model.  The manuscript is well written and the data are convincing. Prior publication I have only some minor comments, which should be addressed: 

  • Line 74: Please write “Hilden” instead of “Hiden”.
  • Figure 2: Please improve the presentation. Right now, the figure is heard to read.
  • Figure 3: Again, please improve the presentation. The data are hard to read. Moreover, the labeling of the y axis is not correct.
  • Lines 270-271: What is the meaning of the sentence: “No neutralizing antibodies were detected in sera of any inoculated piglets.” Please describe the results in more detail. Did you find neutralizing antibodies? Because in “3.6.” (line 272) neutralizing antibodies were used. Please explain.
  • Legend to Figure 5: Please describe the meaning of “Low” and “High”.

Author Response

Response to Reviewer 1 Comments

    We thank the reviewer for helpful comments to construct an improved edition of the manuscript. The experiment contained in this manuscript was designed to explore the effect of deletion of S1 gene for virulence, replication, and antigenicity of PEDV. We feel that this set of data, including figures and table, should be retained as much as possible, as they correlated and complemented to each other. We respectfully request that they remain as it is, for the sake of completeness and for future reference. We believe that, with further understanding of the current issue, some data would be explicable in the future.

Point 1: Line 74: Please write “Hilden” instead of “Hiden”.

Response 1: The "Hiden" has been changed to “Hilden” (Line 77).

Point 2: Figure 2: Please improve the presentation. Right now, the figure is heard to read

Response 2: The Taiwanese non-S-INDEL strains of figure 2 in the original manuscript, non-submitted into NCBI, were replaced by Taiwanese non-S-INDEL strains in GenBank of NCBI. The alignment and bootstrapping were redone. The results of section 3.2 and note of figure 2 was optimized.

Point 3: Figure 3: Again, please improve the presentation. The data are hard to read. Moreover, the labeling of the y axis is not correct.

Response 3: We have edited the labelling of the y axis and marked the major significantly difference in figure 3. The results of section 3.4 and note of figure 3 have been optimized.

Point 4: Lines 270-271: What is the meaning of the sentence: “No neutralizing antibodies were detected in sera of any inoculated piglets.” Please describe the results in more detail. Did you find neutralizing antibodies? Because in “3.6.” (line 272) neutralizing antibodies were used. Please explain.

Response 4:  Because the VN antibodies against PEDV (5-17-O and 5-17-V) were negative in sera of all piglets in section 2.6 experiment at 10 HPI, we collected 66 anti-PEDV sera from the other pig experiments of inactivated 5-17-O and 5-17-V vaccines to do the cross-neutralization between 5-17-O and 5-17-V (section 2.10). The note of figure 5 and section 2.10 and 3.6 have been optimized.

Point 5: Legend to Figure 5: Please describe the meaning of “Low” and “High”.

Response 5: The description of “Low” and “High” mean was added in the note of figure 5.

Reviewer 2 Report

            In this manuscript, “Deletion on S1 Region of Porcine Epidemic Diarrhea Virus Reduces the Virulence and Influences the Virus Neutralizing Activity of the Antibody induced”, Tsai et al. studied two novel PEDV strains (strains 5-17-O and 5-17-V) isolated from Taiwan. They found that the 5-17-V strain had a 205-aa deletion in the S1 region. They conducted experiments to compare the effects of this deletion on viral distribution in herds, virulence, and antigenicity between the two novel PEDV strains. They conclude that the deletion in the S1 region reduces the virulence of PEDV and influences the virus-neutralizing activities induced by this mutant strain. This report presents a solid set of experiments with reasoned results.

Comments:

            In Fig. 1. and Fig. 2, the authors compare the genetic sequences of the two strains and highlight the location of the deletions. However, although the authors list many other viral strains, they should also compare and discuss the differences among them. For example, since the deletion is similar among 5-17-V, LC022792, and KM392229, they should discuss the strains’ origins and behavioral similarities. They should also discuss why they selected these sequences for comparison.

            In Table 1. The authors compare other gene sequences from these two strains and report that the differences reside mainly in the spike gene. Since 5-17-O and 5-17-V are both genotype 2 viruses, they should compare the % difference of these viruses with standard non-S INDEL strains.

            Fig. 3 shows the growth curves of the two viral strains. Previous studies have demonstrated that after trypsin treatment, virus titers increase. The authors should explain why trypsin treatment affects the final titers of 5-17-V but not 5-17-O.

            In Table 2 and Fig. 4, different virus titers are used to infect piglets. The virus shedding time of the full-length S virus strain is longer. The authors found that 50 % mortality occurred in piglets inoculated with 106 TCID50 of 5-17-O (group 4) but 107 TCID50 of 5-17-V (group 8). There are two problems with these experiments. Firstly, the TCID50 concentration should reflect the middle and not final dilution points but, in these experiments, 106 of 5-17-O and 107 of 5-17-V are the final dilution concentrations. Secondly, the authors should explain why the 50% mortality spikes at 4 days post-inoculation with 5-17-V in Fig. 4. 

    Finally, the authors suggest using 5-17-V as a potential vaccine. However, according to their experiments summarized in Fig. 5, the concentration of neutralizing antibodies generated by 5-17-V is low. Therefore, it may not represent a promising vaccine. It would be more reasonable to use a mild strain with a complete spike sequence as a better vaccine.

Author Response

Response to Reviewer 2 Comments

We thank the reviewer for helpful comments to construct an improved edition of the manuscript.

Point 1: In Fig. 1. and Fig. 2, the authors compare the genetic sequences of the two strains and highlight the location of the deletions. However, although the authors list many other viral strains, they should also compare and discuss the differences among them. For example, since the deletion is similar among 5-17-V, LC022792, and KM392229, they should discuss the strains’ origins and behavioral similarities. They should also discuss why they selected these sequences for comparison.

Response 1: In order to understand the sequence differences of strain 5-17-O and 5-17-V from those of non-S-INDEL strains of other countries, variant non-S-INDEL strains, traditional Taiwanese strains, and S-INDEL strain, the sequences of S gene of these strains were aligned and assayed. The results of phylogenetic tree, alignment, and similarity distance were described in sections 3.1 and 3.2. The discussion for analysis of PEDV genome was described between lines 347 and 357.

Point 2: In Table 1. The authors compare other gene sequences from these two strains and report that the differences reside mainly in the spike gene. Since 5-17-O and 5-17-V are both genotype 2 viruses, they should compare the % difference of these viruses with standard non-S INDEL strains.

Response 2: The non-S-INDEL strain (KF452323) and S-INDEL strain (AF353511) were joined to compare with various genes of strain 5-17-V and 5-17-V. The comparison results are list in the table 1 and described in section 3.2.

Point 3: Fig. 3 shows the growth curves of the two viral strains. Previous studies have demonstrated that after trypsin treatment, virus titers increase. The authors should explain why trypsin treatment affects the final titers of 5-17-V but not 5-17-O.

Response 3: The more details about difference of 5-17-O and 5-17-V in the various trypsin concentrations have been described in section 3.4 and figure 3. The effect of trypsin treatment for the 5-17-O and 5-17-V replication is discussed between line 368 and 379.

Point 4: In Table 2 and Fig. 4, different virus titers are used to infect piglets. The virus shedding time of the full-length S virus strain is longer. The authors found that 50 % mortality occurred in piglets inoculated with 106 TCID50 of 5-17-O (group 4) but 107 TCID50 of 5-17-V (group 8). There are two problems with these experiments. Firstly, the TCID50 concentration should reflect the middle and not final dilution points but, in these experiments, 106 of 5-17-O and 107 of 5-17-V are the final dilution concentrations. Secondly, the authors should explain why the 50% mortality spikes at 4 days post-inoculation with 5-17-V in Fig. 4. 

Response 4:

  • The 106 TCID50 /ml of 5-17-O and 107TCID50 /ml of 5-17-V were the highest titers obtained in our lab. Therefore, the study design was to use the highest titer PEDV to serially dilute and challenge the piglets. We determined the titer of the original PEDV before dilution, but not after PEDV dilution.
  • Two piglets in group 4 (5-17-O/106) died separately at 5 and 6 DPI. Two piglets in group 8 (5-17-V/107) died separately at 6 and 8 DPI. Before the piglets’ death, they showed continual severe diarrhea for more than three days. These descriptions have been added to section 3.5.1 (lines 281-283).

Point 5: Finally, the authors suggest using 5-17-V as a potential vaccine. However, according to their experiments summarized in Fig. 5, the concentration of neutralizing antibodies generated by 5-17-V is low. Therefore, it may not represent a promising vaccine. It would be more reasonable to use a mild strain with a complete spike sequence as a better vaccine.

Response 5: In lines 385-389, the sentence “A low-virulence PEDV strain, such as 5-17-V, can be a potential candidate for a vaccine strain” was edited to “The variant non-S-INDEL (5-17-V), a low-virulent PEDV strain, has the advantage of easier attenuation, but its antigenicity to induce the antibody production is not completely characterized. Therefore, it is not a good candidate for vaccine development. However, a mild virulence strain with a complete S protein may be a better choice for vaccine candidates.”.

Reviewer 3 Report

The manuscript describes the differences between two PEDV strains, 5-17-O and 5-17-V. The researcher compared the two isolates for nucleotide and protein differences, as well as in vitro replication and pathogenicity. 5-17-V has a large aa deletion and 3 other aa differences when the two S-gene proteins are evaluated. In vitro replication analysis under several concentration of trypsin suggested that they were equal when evaluated at 5 and 10 ug/ml trypsin, but differ at lower concentrations. In vivo analysis compared the two isolates at 3-7 logs of 5-17-O and 4-7 logs of 5-17-V. At lower titers, 5-17-V produced lower average days of PEDV shedding in feces, but higher average days of PEDV shedding in feces at higher titers. Anti-neutralizing antibodies were also examined – the data suggests there was a problem with the analysis.

Because of uncertainties with the antibody analysis and problems with other aspects of the manuscript, the actual claim suggested by the title was no supported by clear evidence.

Specific comments:

  1. Line 34: “The PED virus (PEDV) is an enveloped, single-stranded, positive-sense RNA virus with the insertion of a spike (S) protein on the envelope.” Incorrect phrasing. The spike protein and other proteins are included in the envelope of PEDV; it is not a PEDV virus with an insertion of S. Rephrase.
  2. Line 151: Do you mean 5-17-O?
  3. Do not see GenBank Accession numbers for the genomes of 5-17-O and 5-17-V. Add to manuscript.
  4. Line 170: Were other silent mutations noted between the two strains or in noncoding regions? If so, they must be added to provide absolute clarity on the differences between the two genomes.
  5. When assessing Figure 1, I saw that aa 254 was different between the two PEDV strains. Make sure you are correct on all of the changes in the S gene. Perhaps use Fig. 1 alignment numbers to show where the differences are, rather than just an alignment of the two 5-17 genomes.
  6. Line 196: “1-aa insertion between positions 135 and 136”. Fig. 1 shows the insertion to be between 139 and 149, which includes the differences noted above. Make certain you are clear to the readers and use numbering in Fig. 1 to describe differences.
  7. 3: Increase fonts for the different symbols. Very hard to read.
  8. Line 271: Add “data not shown”
  9. Line 273: At what time post-infection was the serum examined for neutralizing antibodies? Add to manuscript. Fig. 5: Significant differences seem off as the error bars suggest all were not significant. Review and speak to a mathematition.
  10. Many incorrect phrases and grammar mistakes were noted throughout the manuscript. Recommend commercial editing service.

Author Response

Response to Reviewer 3 Comments

We thank the reviewer for helpful comments to construct an improved edition of the manuscript.

Point 1: The manuscript describes the differences between two PEDV strains, 5-17-O and 5-17-V. The researcher compared the two isolates for nucleotide and protein differences, as well as in vitro replication and pathogenicity. 5-17-V has a large aa deletion and 3 other aa differences when the two S-gene proteins are evaluated. In vitro replication analysis under several concentration of trypsin suggested that they were equal when evaluated at 5 and 10 ug/ml trypsin, but differ at lower concentrations. In vivo analysis compared the two isolates at 3-7 logs of 5-17-O and 4-7 logs of 5-17-V. At lower titers, 5-17-V produced lower average days of PEDV shedding in feces, but higher average days of PEDV shedding in feces at higher titers. Anti-neutralizing antibodies were also examined – the data suggests there was a problem with the analysis.

Because of uncertainties with the antibody analysis and problems with other aspects of the manuscript, the actual claim suggested by the title was no supported by clear evidence

Response 1: Because the description of cross-neutralization between 5-17-O and 5-17-V was not clear, it led that the results of cross-neutralization experiment was confused with the results of virus neutralizing assay in the virulence experiment. We have optimized the section 2.10 of materials and methods, section 3.6 of results, and note of figure 5 in the revised manuscript. In the experiment of cross-neutralization, because the VN antibodies against PEDV (5-17-O and 5-17-V) were negative in sera of all piglets in the section 2.6 experiment at 10 HPI, we collected 66 anti-PEDV sera from the other pig experiments of inactivated 5-17-O and 5-17-V vaccines to do cross-neutralization between 5-17-O and 5-17-V (section 2.10).

Point 2: Line 34: “The PED virus (PEDV) is an enveloped, single-stranded, positive-sense RNA virus with the insertion of a spike (S) protein on the envelope.” Incorrect phrasing. The spike protein and other proteins are included in the envelope of PEDV; it is not a PEDV virus with an insertion of S. Rephrase.

Response 2: The sentence “The PED virus (PEDV) is an enveloped, single-stranded, positive-sense RNA virus with the insertion of a spike (S) protein on the envelope“ has been edited to” The PED virus (PEDV) is an enveloped, single-stranded, positive-sense RNA virus with the spike (S) protein on the envelope” (Lines: 36-37).

Point 3:    Line 151: Do you mean 5-17-O?

Response 3: Yes, the “…5-17-V or 5-17-V…” was edited to “…5-17-V or 5-17-O…” (line 158)

Point 4: Do not see GenBank Accession numbers for the genomes of 5-17-O and 5-17-V. Add to manuscript.

Response 4: The accession number has been added to section 2.2 (line 88-89).

Point 5: Line 170: Were other silent mutations noted between the two strains or in noncoding regions? If so, they must be added to provide absolute clarity on the differences between the two genomes.

Response 5: Comparison of the sequences of 5’UTR and 3’UTR between novel Taiwanese strains and reference strains are described in sections 3.1, 3.2, and table 1

Point 6: When assessing Figure 1, I saw that aa 254 was different between the two PEDV strains. Make sure you are correct on all of the changes in the S gene. Perhaps use Fig. 1 alignment numbers to show where the differences are, rather than just an alignment of the two 5-17 genomes.

Response 6: The three changes from the 5-17-O to the 5-17-V strain were from S to G, from P to L, and from P to Q in aligned positions 254, 506, and 1063, respectively. In figure 1, we mark the insertion and deletion between non-S-INDEL and S-INDEL strains, the difference between 5-17-O and 5-17-V, and the same amino acids among all PEDV strains. The alignment numbers have been used in the figure 1 and section 3.1of results.

Point 7: Line 196: “1-aa insertion between positions 135 and 136”. Fig. 1 shows the insertion to be between 139 and 149, which includes the differences noted above. Make certain you are clear to the readers and use numbering in Fig. 1 to describe differences

Response 7: We used the alignment number to describe the insertion, deletion, and difference of PEDV strains in sections 3.1, 3.2, and figure 1.

Point 8: 3Increase fonts for the different symbols. Very hard to read.

Response 8: We have increased the fonts and marked the major significantly differences in figure 3. The results of section 3.4 and note of figure 3 have been optimized.

Point 9: Line 271: Add “data not shown

Response 9: We have added it to line 312.

Point 10: Line 273: At what time post-infection was the serum examined for neutralizing antibodies? Add to manuscript. Fig. 5: Significant differences seem off as the error bars suggest all were not significant. Review and speak to a mathematition.

Response 10: The VN antibodies against PEDV (5-17-O and 5-17-V) were negative in the sera of all piglets in the section 2.6 experiment at 10 HPI. To understand the cross-neutralization between 5-17-O and 5-17-V, we collected 66 anti-PEDV sera from the other pig experiments of inactivated 5-17-O and 5-17-V vaccines to assay the difference of VN titers against 5-17-O and 5-17-V. The VN titers against the strain 5-17-V in sera of the 5-17-O low, 5-17-V low, and 5-17-V high groups were all equal/similar or higher than that against strain 5-17-O. The ratio of the VN titers against the 5-17-V strain were over four-fold higher than that of the VN titer against the 5-17-O strain: 30.4% (7/23), 47.1 % (8/17), and 72.7% (8/11) in the 5-17-O low; 5-17-V low and 5-17-V high, respectively. The neutralizing titers against 5-17-V strains in the 5-17-O low (5.0 ± 2.3 log2), 5-17-V low (2.9 ± 1.6 log2), and 5-17-V high (6.7 ± 1.0 log2) samples were,  significantly higher than that against 5-17-O (3.6.0 ± 1.1 log2 of 5-17-O low; 2.9 ± 1.6 log2 of 5-17-V low; 4.1 ± 1.6 log2 of 5-17-V high, respectively). In addition, the mathematics of results is described in the section 3.6.

Point 11: Many incorrect phrases and grammar mistakes were noted throughout the manuscript. Recommend commercial editing service.

Response 11: We agree with the reviewer’s assessment. In order to make this manuscript fluent and easy to read, it is proofed by the English Editing Services of “MDPI Author Services” to assist in optimization.